# Construction of HER2-Specific HIV-1-Based VLPs

**DOI:** 10.3390/bioengineering9110713

**Published:** 2022-11-19

**Authors:** Sofia A. Martins, Joana Santos, Sandra Cabo Verde, João D. G. Correia, Rita Melo

**Affiliations:** 1Centro de Ciências e Tecnologias Nucleares, Instituto Superior Técnico, Universidade de Lisboa, Estrada Nacional 10, ao km 139,7, 2695-066 Loures, Portugal; 2Departamento de Engenharia e Ciências Nucleares, Instituto Superior Técnico, Universidade de Lisboa, Estrada Nacional 10, ao km 139,7, 2695-066 Loures, Portugal

**Keywords:** HIV-1-based virus-like particles, HEK293T cells, single-chain variable fragments, HER2, targeted therapy

## Abstract

Virus-like particles (VLPs) are nanoplatforms comprised of one or more viral proteins with the capacity to self-assemble without viral genetic material. VLPs arise as promising nanoparticles (NPs) that can be exploited as vaccines, as drug delivery vehicles or as carriers of imaging agents. Engineered antibody constructs, namely single-chain variable fragments (scFv), have been explored as relevant molecules to direct NPs to their target. A vector containing the scFv of an antibody, aimed at the human epidermal growth factor receptor 2 (HER2) and fused to the human immunodeficiency virus (HIV) protein gp41, was previously constructed. The work herein describes the early results concerning the production and the characterization of HIV-1-based VLPs expressing this protein, which could function as potential non-toxic tools for transporting drugs and/or imaging agents.

## 1. Introduction

Virus-like particles (VLPs) are widely recognized for their immunogenic features but their flexibility allows them to be employed for a myriad of applications [1]. Lately, they are at the forefront of research into vaccinations, gene therapy and drug delivery [2]. A major feature of VLPs is their innate capacity to reach specific targets and deliver their cargo in a strictly controlled mode. VLPs are derived from non-replicating viruses (lacking the viral genome), which can be defined as soft-matter biological structures that can be modified by genetic means or chemical modification [3,4]. Furthermore, their stability in biological fluids and their lack of interaction with mucosal tissues, serum or plasma proteins, indicate that these nanoplatforms may be particularly fitting for theragnostic applications and that the virus capsid does not affect their targeting properties [5]. One strategy that has been proposed to improve the delivery of cell-specific drugs and/or imaging agents relies on the expression of antibodies on the surface of the viral membrane that recognizes a cell-type specific receptor [6]. In cancer treatment, targeted VLPs become even more attractive as they harness the enhanced permeability and retention (EPR) effect for selective accumulation in tumors [7]. VLPs can also help overcome other shortcomings associated with standard drugs, namely poor solubility, low bioavailability and/or unfavorable pharmacokinetics. In addition, multidrug resistance (MDR), provoked by drug efflux transporters, which are frequently overexpressed in cancer cells, can be addressed by a specific delivery [8].

Within the wide range of VLPs that are currently under development, HIV-1-based VLPs have been gaining relevance due to their pertinence as vaccine candidates [9,10]. The Gag polyprotein, one of the HIV constituents, can form VLPs without any other viral component, enhancing the attractiveness of this VLP type [11,12].

In previous work, we successfully constructed a plamid harboring a scFv of trastuzumab, a well-characterized anti-human epidermal growth factor receptor 2 (HER2) antibody, as a targeting moiety fused to the HIV viral protein gp41 and tagged with hemagglutinin (HA) [13]. The validation was performed through transient transfection of the vector [13]. Here, we took advantage of this previous work and went on to construct HER2-specific HIV-1-based VLPs (Figure 1) and proposed a new strategy to overcome the limitations associated with the current HIV-1-based VLP production systems [14]. Using a mammalian expression system, we tested different ratios of total DNA, the number of cells and the time of collection of the VLPs post transfection [15,16]. These three factors are key to the successful production of VLPs and are often overlooked to the detriment of the type of transfection system [17]. The production, morphology and cytotoxicity of HIV-1 VLP were carefully evaluated. This brief research approach may provide novel insights into strategies for the development of targeted therapy tools that can harness recombinant proteins to be expressed in VLPs.

## 2. Materials and Methods

### 2.1. Plasmid Construction and Transformation

The pX1665 plasmid, containing the anti-HER2 scFv and the HIV protein gp41, was synthesized by Synbio Tech. (South Brunswick, NJ, USA). The pMDLg/pRRE plasmid (Addgene, Watertown, MA, USA) codes for the Gag, Pol and Rev response element (RRE). The pRSV-REV plasmid (Addgene, USA) encodes the Rev protein. The plasmids were replicated in NEB^®^ Stable Competent Escherichia coli, and the cells were transformed by electroporation, in accordance with the protocol described by Lessard [18].

### 2.2. Cell Culture 

The HEK-293T (human embryonic kidney comprising 293 cells that express the large T antigen of the simian virus 40), SK-BR-3 (HER2-overexpressing breast cancer cells) and MDA-MB-231 (breast cancer cells that lack HER2 overexpression) cell lines were obtained from the American Type Culture Collection (ATCC, Manassas, VA, USA). Cell cultures was carried out in Dulbecco’s Modified Eagle’s Medium (DMEM)/High Glucose, supplemented with 10% (*v/v*) fetal bovine serum (FBS), 10 mM HEPES, 1× MEM non-essential amino acids, 1 mM sodium pyruvate and 1% penicillin/streptomycin (all reagents were purchased from Cytiva, (Marlborough, MA, USA). The cells were maintained in 75 cm^3^ disposable polycarbonate cell culture flasks at 37 °C in a humidified incubator with 5% CO_2_. 

### 2.3. Transfection 

For transfection, two protocols were performed. In the first protocol, HEK-293T cells were plated onto 10 cm plates at a density of 7.0 × 10^6^ cells/plate and transfected with Lipofectamine 3000 (Cat# L3000001, Thermo Fisher Scientific, Waltham, MA, USA) as per the manufacturer’s protocol, using 17.5 μg of pX1665 and 21.0 μg of pMDLg/pRRE (1:1 ratio). Cells were incubated at 37 °C for 6 h with 5% CO_2_, and the medium was then discarded and replaced with fresh complete medium. The supernatant was retrieved 24 h post-transfection, and 12 mL of fresh medium were added. At 52 h post-transfection, the supernatant was retrieved again (Figure 2A) [15]. Both collections were combined and centrifuged at 2000 rpm (Beckman, J2-21M, Beckman Coulter, Inc., Brea, CA, USA) at room temperature for 10 min. In the second protocol, HEK-293T cells were plated into 6-well plates at a density of 1.0 × 10^6^ cells/well and transfected with Lipofectamine 3000 (Cat# L3000001, Thermo Fisher Scientific, Waltham, MA, USA) as per the manufacturer’s protocol, using different ratios of pX1665, pMDLg/pRRE and pRSV-REV, depicted in Table 1. Cells were incubated at 37 °C for 6 h with 5% CO_2_, and the medium was then discarded and replaced with fresh complete medium. At 48 h post-transfection, the supernatant was collected and centrifuged for 10 min at 2000 rpm at room temperature (Figure 2B) [16]. 

### 2.4. Enzyme-Linked Immunosorbent Assay

Quantification of the HIV-1-based VLPs was conducted using the Enzyme-Linked Immunosorbent Assay (ELISA) INNOTEST HIV Antigen mAb (Fujirebio, Tokyo, Japan) in accordance with the manufacturer’s protocol. The absorbance at 450 nm was measured on an EZ Read 800 Microplate Reader (Biochrom, Holliston, MA, USA) and quantified against a p24 standard curve. The number of VLPs was estimated through the empirical determination that 1 pg of p24 corresponds to around 1 × 10^4^ particles [19]. 

### 2.5. Western Blot

The collected samples were mixed with 1/6 of a 6× sodium dodecyl sulfate (SDS) sample loading buffer and heated at 95 °C for 10 min. The samples were loaded onto a 12.5% SDS-polyacrylamide gel and separated by electrophoresis. Following separation, the samples were transferred onto a nitrocellulose membrane (BioRad, Hercules, CA, USA). The membrane was blocked at 4 °C overnight with 5% (*w/v*) non-fat dried milk in phosphate buffered saline (PBS) containing 0.2% (*v/v*) Tween 20 (PBS-T). The membrane was washed with PBS-T and incubated with an anti-HA monoclonal antibody (Cat# 901502, BioLegend, San Diego, CA, USA) diluted 1:2000 in 1% (*w/v*) non-fat dried milk in PBS-T for 60 min at room temperature with a gentle agitation. The membrane was then washed with PBS-T and incubated with an anti-mouse secondary antibody (BioRad, USA) diluted 1:3000 in 1% (*w/v*) non-fat dried milk in PBS-T for 60 min at room temperature with a gentle agitation. The membrane was then washed with PBS-T. Visualization of the proteins was carried out with the ECL^®^ reagent (GE Lifesciences, Chicago, IL, USA) in accordance with the manufacturer’s protocol.

### 2.6. Transmission Electron Microscopy

Transmission electron microscopy (TEM) of the collected VLPs was performed at the Instituto Gulbenkian de Ciência—Electron Microscopy Facility. The collected VLPs were first precipitated and concentrated using the Lenti-X Concentrator (Cat# 631231, Takara Bio, Shiga, Japan) as per the manufacturer’s protocol and resuspended in PBS to minimize cell debris and impurities in the TEM images. Briefly, 5 μl of concentrated VLP suspensions were placed on formvar/carbon-coated glow-discharged copper EM grids and absorbed for 5 min. The grids were washed 10 times with distilled water and were then stained with one drop of 2% uranyl acetate for 5 min. TEM analysis was conducted with a FEI Tecnai G2 Spirit BIOTWIN microscope at 120 keV. Images were captured using an Olympus-SIS Veleta CCD Camera at 43K and processed using ImageJ software [20].

### 2.7. Cytotoxicity Assay

Cell viability was assessed through the tetrazolium salt WST-1 (4-[3-(4-iodophenyl)-2-(4-nitrophenyl)-2H-5-tetrazolio]-1,3-benzene disulfonate) assay. SK-BR-3 and MDA-MB-231 cells were plated at a density of 1.0 × 10^5^ cells/well in 96-well plates and incubated at 37 °C and 5% CO_2_ for 52 h. 1/10 of the concentrated VLPs were added to the cells. After 24 h of incubation at 37 °C, the inoculum was removed from the wells and 100 μL of fresh medium (DMEM 10% FBS) and 10 μl of the cell proliferation WST-1 reagent (Roche, Switzerland) were added. Following gentle mixing, the cells were incubated at 37 °C for 6 h. The absorbance was measured on an EZ Read 800 Microplate Reader (Biochrom, USA) at 450 nm, using a reference wavelength of 620 nm. The results were depicted as percentages of cell viability of the treated cells with reference to the untreated cells. The values were presented as the mean of the triplicates with error bars representing the 95% confidence interval for the mean. Data were statistically analyzed by one-way analysis of variance (ANOVA), at a significance level of *p* < 0.05.

## 3. Results

### 3.1. Production and Optimization of HER2-Specific HIV-1-Based VLPs

A previously constructed biomimetic vector [13], termed pX1665, was used in this study. Aimed at investigating which variables would affect the VLP assembly, we produced HER2-specific HIV-1-based VLPs in HEK-293T cells, grown in adherent cultures by transient transfection, using two protocols. In the first protocol, HEK293T cells were plated at a density of 7.0 × 10^6^ cells/plate on 10 cm plates, transiently transfected with pX1665 and pMDLg/pRRE at a 1:1 ratio and the VLP collection occurred at 24 h and 52 h post-transfection, as per the manufacturer’s protocol for lentiviral production [15]. The obtained yield was low, and it was observed that VLP collection at 24 h post-transfection disrupted cell viability and consequently decreased the VLP production. A single generation of VLPs, at 48 h post-transfection was thus selected for the second protocol, given that it had already been described in the literature [16]. Furthermore, pRSV-REV, a plasmid encoding *rev*, was added to the transfection protocol due to the relevance of the Rev protein in the transcription of structural genes [14]. The number of cells was reduced as high cell densities significantly decreased transfection efficiency [14]. Lastly, different ratios of the plasmid, containing the anti-HER2 scFv, were tested to ascertain whether they affect the incorporation of the recombination protein and the VLP assembly. The second protocol thus includes plating of HEK-293T at a density of 1.0 × 10^6^ cells/well into 6-well plates, transient transfection with pX1665, pMDLg/pRRE and pRSV-REV at 1:1:1, 2:1:1, 5:1:1 and 10:1:1 ratios with the VLP collection taking place at 48 h post-transfection (Figure 2). 

The validation of the constructed VLPs and the determination of p24 concentration was accomplished by ELISA. Optimization of the production of the HIV-1-based VLPs addressed the ratio of the total DNA, transfected cell conditions and the time of the supernatant recovery. The yields obtained through the different protocols were compared, and the highest number of VLPs (1.41 × 10^7^ VLPs/mL) was attained via crude harvested supernatants using a ratio of 1:1:1 with 1.0 × 10^6^ cells/well plated into the 6-well plate with collection at 48 h post-transfection (Figure 3). This protocol for retargeted VLP production serves as a starting point to better understand the behavior of these VLPs under basic production conditions, providing a simple and feasible workflow. From here on, we will consider the concentration and purification of these VLPs in order to make this system more profitable and a scale-up achievable.

### 3.2. Production and Optimization of HER2-Specific HIV-1-Based VLPs

Following the production and the optimization of HER2-specific HIV-1-based VLPs, the Western Blot method was performed to determine whether the anti-HER2 scFv is expressed in the constructed VLPs. Figure 4A depicts the Western Blot analysis conducted with an anti-HA antibody, and the presence of the anti-HER2 scFv in the constructed VLPs was confirmed.

The molecular weight of the fusion protein is about 49 kDa (Figure 4, lanes 3, 4, 5, 6, 7 and 8), which is slightly higher than the previously determined theoretical value (47.1 kDa) [13]. This validates the presence of the anti-HER2 scFv in the VLPs. After verifying that the protein of interest is expressed in the VLP, we proceeded to the morphological characterization of the VLPs to understand if the VLPs were assembled correctly. Figure 4B,C show spherical structures that display the morphology associated with negative stained HIV-1 VLPs [21,22]. The average size of the VLPs was obtained through ImageJ software. HIV-1 Gag VLPs (Figure 4B) display an average size of 114.17 ± 20.36 nm, which lies within the characteristic range of HIV-1 VLPs (100–200 nm), and is in accordance with previously determined diameters for these particles [23].The constructed HER2-specific HIV-1-based VLPs display an average size of 132.09 ± 12.52 nm, which agrees with the described diameters for wild-type HIV-1 VLPs [23], and indicates that the incorporation of the scFv-HER2_gp41 recombinant protein does not alter the size or morphology of VLPs.

Lastly, a cytotoxicity assay was conducted to assess the antiproliferative properties of the constructed VLPs in HER2-negative and HER2-positive cancer cell lines (Figure 5). Two breast cancer cell lines, MDA-MB-231 (lacking HER2 overexpression) and SK-BR-3 (HER2-overexpressing) were thus treated with concentrated VLPs, diluted at 1/10. Statistical analysis shows that there is no significant difference between the cell viability of treated cells when compared with the cell viability of untreated cells. The results obtained suggest that the constructed VLPs do not appear to be cytotoxic to either breast cancer cells that lack HER2 overexpression and breast cancers that display HER2 overexpression, as there was no substantial decrease in cell viability. Notwithstanding, given the proof-of-concept nature of this study, the tested concentrations are small. A scale-up and further tests are thus required. Such results validate our retargeted VLPs as potential non-toxic tools for transporting drugs and/or imaging agents.

## 4. Discussion

The growing necessity to develop efficient therapies for diseases, such as cancer has spurred the design of novel nanoplatforms aimed at targeted therapy. VLPs are nanoscale platforms comprised of assembled viral proteins, which lack the viral genome, have emerged as an alternative to conventional nanoparticles because of their biocompatibility, stability, and their ability to target certain cells and tissues in order to deliver their cargo in a precise manner [2,3,4]. VLPs are mainly harnessed for vaccine development [24], but their versatility allows them to be used as drug delivery vessels and/or carriers of imaging agents [6]. 

Among the different types of VLPs, HIV-1-based VLPs are particularly promising because self-assembly can be achieved with just one polyprotein [11]. The ability of this type of VLP to boost strong immune reactions prompted different studies to report upon their applicability as vaccine candidates [10,25,26]. Despite their evident applicability as vaccine candidates, HIV-1-based VLPs can also be employed as delivery vehicles due to their capacity to be engineered and to encapsulate drugs [27,28,29]. Notwithstanding, the establishment of HIV-1-based VLPs as therapeutic nanoplatforms is still immature. There are several shortcomings that need to be taken into consideration when designing HIV-1-based VLPs, namely the formation of mature virions, which may carry genetic material and have deleterious effects [30]. The production process, which can involve contamination with cellular components, such as exosomes, microvesicles and RNA can hinder the medical applications of these VLPs [31]; moreover, expression platforms, which must be carefully selected to assure an adequate glycosylation pattern can influences VLP safety, stability and efficacy [32]. 

In this work, we aimed to design HIV-1-based VLPs that are directed at HER2 because of its well-established role in cancers, such as breast cancer [33] and gastric cancer [34]. We focused on the production process and its optimization at a laboratory scale, and used a mammalian expression system, the HEK-293T cell line, as it is easy to handle [35] and confers complex post-translational modifications [14]. Furthermore, we performed preliminary assays that may pave the way for the establishment of this nanostructure as a targeted therapy platform. 

We used a previously constructed vector containing the scFv of an anti-HER2 antibody, trastuzumab, fused to the HIV viral protein gp41 [13] for co-transfection with pMDLg/pRRE, which encodes the HIV structural polyprotein Gag, the polyprotein precursor Pol and RRE, and pRSV-REV, which in turn encodes the accessory protein Rev. Following plasmid replication in *E. coli*, HEK-293T cells were seeded at different cell densities and transiently transfected with different plasmid ratios; the VLPs were then harvested at different times post-transfection for optimization. It was demonstrated that a density of 1.0 × 10^6^ cells/well in 6-well plates with a 1:1:1 plasmid ratio and a VLP collection at 48 h post-transfection were the optimal conditions to obtain the highest HER2-specific HIV-1-based VLP yields in vitro. The technology herein described serves as a starting point to produce these VLPs and prompts further studies, which are focused on both scalability and cost-effectiveness.

After establishing the protocol for VLP production, we proceeded to the characterization of the constructed VLPs to assess whether the produced VLPs were correctly assembled and displayed the expected characteristics. We started by performing a Western Blot to evaluate the presence of the protein of interest. The expected molecular weight of the fusion protein, 47.1 kDa, was previously determined [13], and the obtained molecular weight, 50 kDa, was similar, confirming the presence of the protein of interest on the VLPs. Next, the constructed VLPs were morphologically characterized through TEM analysis to determine whether the VLPs could assemble correctly. The TEM images show that the constructed VLPs display the expected morphology [21,22]. Moreover, the diameters of the VLPs were measured, and they ranged from 100 to 200 nm, which is in agreement with the literature [23]. Lastly, a cytotoxicity assay was conducted to assess the antiproliferative properties of the constructed VLPs in breast cancer cells that overexpress HER2 and triple negative breast cancer cells that lack such an overexpression. The main conclusion drawn is that the cell viability of both cell lines was not significantly altered, suggesting that these VLPs may not be cytotoxic and could thus be potentially used as a target-specific delivery platform of drugs and/or diagnostic imaging agents for cancer theranostics. These results indicate that our protocol can successfully produce functional VLPs that can be potentially employed for drug delivery and imaging. Further studies regarding large-scale production, binding affinity and in vivo effects are required; however, this work serves as a starting point for such studies.

The goal of this study was accomplished as the innovative VLP was successfully constructed. The research of VLPs in precise medicine is growing exponentially. Therefore, we believe that this fast and systematic approach to the production of retargeted VLPs can have a positive impact in the scientific community.

## Figures and Tables

**Figure 1 bioengineering-09-00713-f001:**
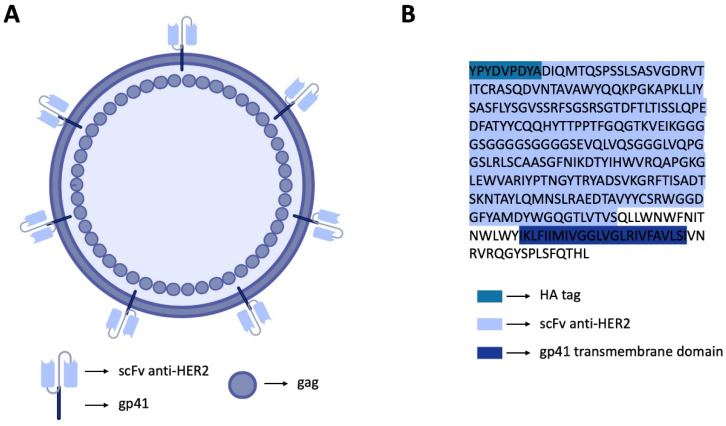
Schematic illustration of the HER2-specific HIV-1-based VLP and the constructed biomimetic vector. (**A**) Illustration of the HER2-specific HIV-1-based VLP. (**B**) Amino acid sequence of the protein comprising the anti-HER2 scFv fused to HIV gp41, with a HA tag at the N-terminal.

**Figure 2 bioengineering-09-00713-f002:**
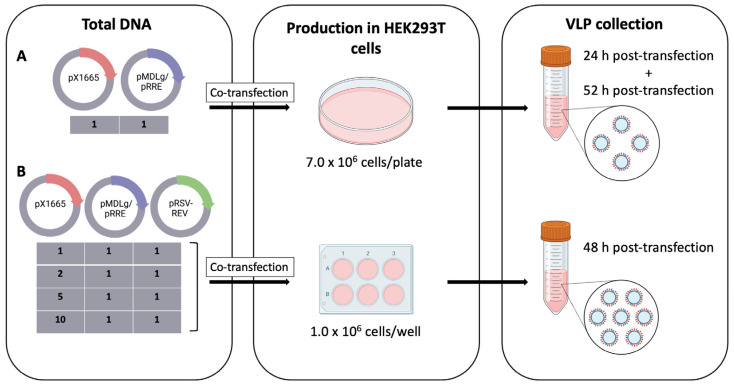
Schematic outline of the production of HER2-specific HIV-1-based VLPs. (**A**) Transfection of HEK-293T cells plated at 7.0 × 10^6^ cells/plate in 10 cm plates with pX1665 and pMDLg/pRRE at a 1:1 ratio and VLP collection at 24 h post-transfection and 52 h post-transfection. (**B**) Transfection of HEK-293T cells plated at 1.0 × 10^6^ cells/well into 6-well plates with pX1665, pMDLg/pRRE and pRSV-REV at different ratios and VLP collection at 48 h post-transfection.

**Figure 3 bioengineering-09-00713-f003:**
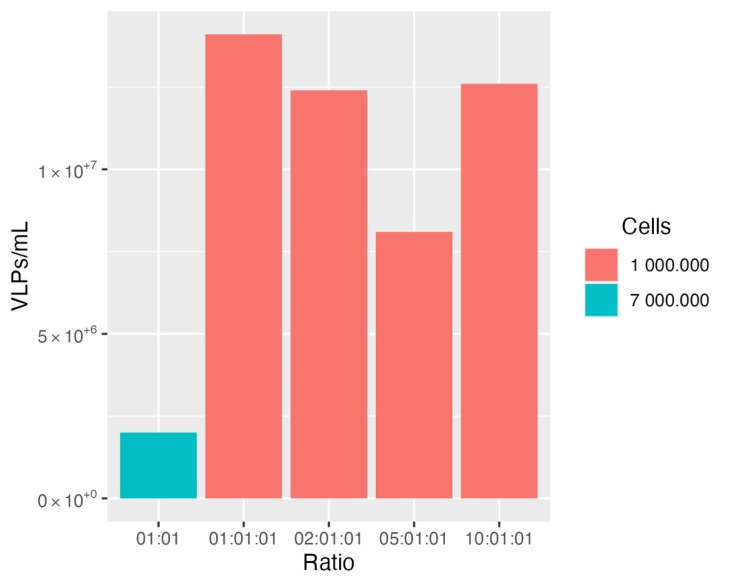
Comparison of different conditions to produce HER2-specific HIV-1-based VLPs. The highest concentration (VLPs/mL) was obtained using a 1:1:1 ratio with 1.0 × 10^6^ cells/well seeded in 6-well plates and collection at 48 h post-transfection.

**Figure 4 bioengineering-09-00713-f004:**
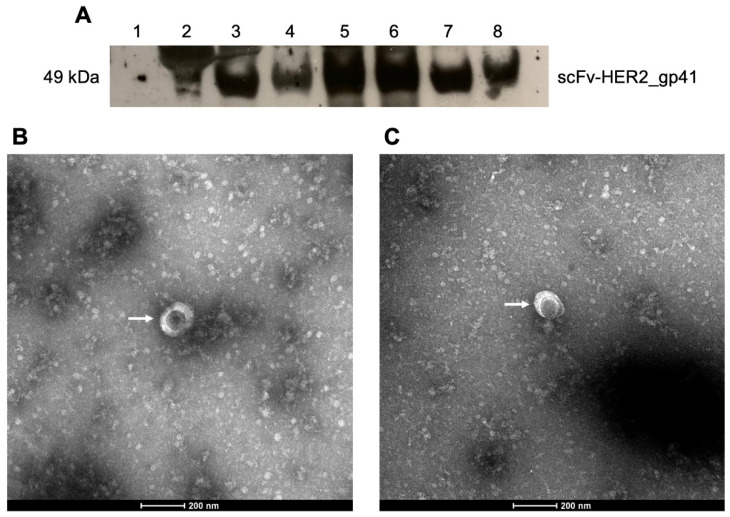
Characterization of HER2-specific HIV-1-based VLPs produced with pX1665, pMDLg/pRRE and pRSV-REV in HEK-293T cells plated at a density of 1.0 × 10^6^ cells/well on 6-well plates and collected 48 h post-transfection. (**A**) Western Blot of the VLPs. Lane 1: negative control (extract lacking HA); Lane 2: positive control (extract containing HA); Lanes 3, 4, 5, 6, 7 and 8: Different VLP batches. Western Blot detection was performed using an anti-HA monoclonal antibody (1:2000) and an anti-mouse secondary antibody (1:3000). Visualization of the bands was performed with the ECL^®^ reagent. All the VLP batches display the HA-tagged scFv-HER2_gp41 fusion protein. (**B**) TEM image of a HIV-1 Gag VLP (depicted by a white arrow). Scale bar = 200 nm. (**C**) TEM image of a HER2-specific HIV-1 VLP (depicted by a white arrow). Scale bar = 200 nm.

**Figure 5 bioengineering-09-00713-f005:**
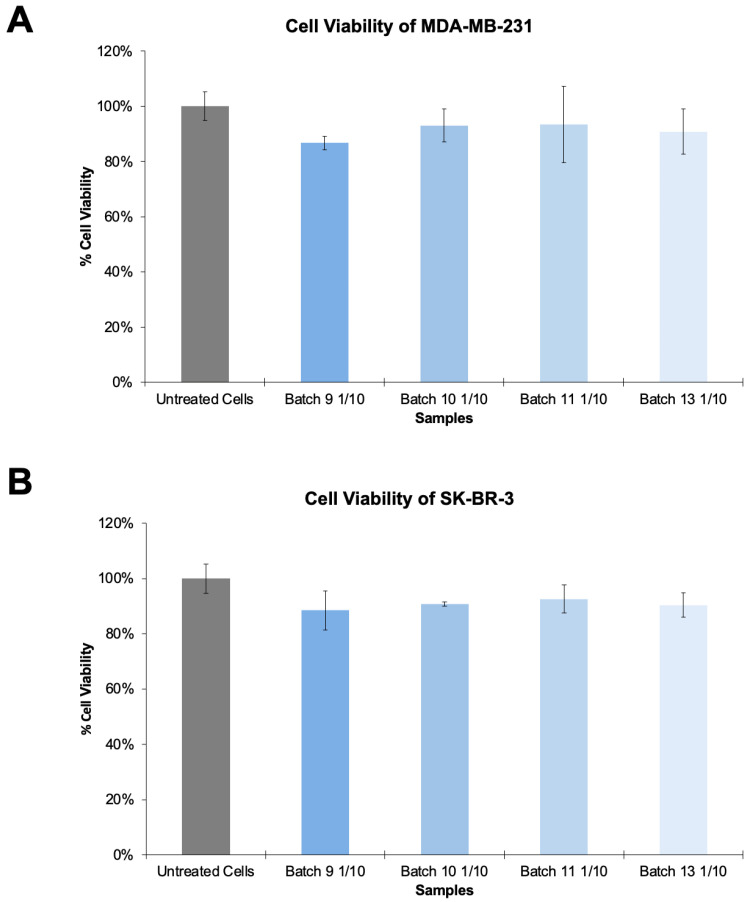
Cytotoxic evaluation of HER2-specific HIV-1-based VLPs. Cytotoxicity assays performed in triplicates in breast cancer cells that lack HER2 overexpression and HER2-overexpressing breast cancer cells. (**A**) Cytotoxicity assay in MDA-MB-231 cells (lacking HER2 overexpression). (**B**) Cytotoxicity assay in SK-BR-3 cells (HER2-overexpressing).

**Table 1 bioengineering-09-00713-t001:** Different plasmid ratios used for transfection of HEK-293T cells in 6-well plates.

pX1665: pMDLg/pRRE: pRSV-REV Ratio	Mass (μg)		
	pX1665	pMDLg/pRRE	pRSV-REV
1:1:1	1.02	1.35	0.63
2:1:1	1.52	1.01	0.47
5:1:1	2.16	0.57	0.27
10:1:1	2.51	0.33	0.16

## Data Availability

Not applicable.

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
