# Peer review of "Construction of HER2-Specific HIV-1-Based VLPs"

_bioengineering, 2022, doi:10.3390/bioengineering9110713_

Round 1

Reviewer 1 Report

Nowadays, using alternative ways to target cells is a hot topic. 

This study which presents the production and characterization of HIV-1-based VLPs expressing this protein, as potential non-toxic tools for drug delivery and images agent, is a relevant proof of concept for further investigations.

The introduction section clearly reports the state of the art and the methodology with respect to state of the art, supported by a schematic figure that describes the system.

In the methods section, the protocols used for the development and characterization are detailing reported.

The results are clearly presented and supported by figures. 

The quality of figure 3 needs to be improved. On the other hand, the figures’ caption well describes the figure panel. 

However, the author has not considered reporting a negative control for the cytotoxicity assay.

Have the Authors tested the system without trastuzumab?

The author should insert the appropriate reference in lines 54, 90,98, 154, 184, 193.

The conclusion is in agreement with the results reported.

More in general the paper is well written and easy to read.

I’m glad to consider this manuscript for publication with minor revision.

Reviewer 2 Report

In this work, HER2-Specific HIV-1-Based VLPs are constructed. The reviewer thinks this paper is interesting both for academia and biopharmaceutical industry. On the other hand, there are several unclear points. The major comments are as follows:

·         In general, the paper is very short and the amount of data that is presented is low. The most important issue is that no HIV particle concentrations are ever mentioned. In line 167 you write that VLP concentration was determined. There is only a reference of ng protein/mL but how is the conversion? What is the number of particles in your other transfections you only mention the highest concentration observed?

·         I suggest adding a plot that shows VLP concentrations obtained in all experiments, to help the reader understand the effects of different ratios of total DNA, number of cells and time of collection of VLPs post transfection

·         How do you assure that lack of cytotoxicity is not an artifact of (too) low VLP concentration?

·         For confirming the morphology, you apply TEM. By identifying four potential particles, you conclude that the morphology is correct. Can you support this assumption by additional data, e.g., from MALS/DLS, etc.? How does it compare to published diameters for HIV1-derived particles?

·         In line 184 you argue that the fusion protein has a MW of 50 kDa, but your bands appear well above 55 kDa, please clarify

·         Section 2.6, and, e.g., line 195 what does concentrated mean? Please quantify. What was the concentration factor? What was the concentration VLPs (e.g. in particles/mL) before and after concentration step?

·         What was the rationale behind the harvesting time you applied? Could you have harvested later to obtain higher VLP concentrations?

·         References are incorrect

·         Check your exponents, for example in line 85/93 and others

·         Is the style of your figure captions from the MDPI Template? Line separation and font size is very large.

·         Did you purify your VLPs before your cytotoxicity assay?

In conclusion, the work is interesting but a more solid and quantitative presentation of the data is needed. I therefore recommend accepting the paper only after further major revision and especially after adding the VLP concentration for each of the conducted transfection reactions.

Round 2

Reviewer 2 Report

The main points of review have been dealt with in sufficient form, I recommend to accept the article after further minor corrections.